# Fusion iENA Scholar Study: Sensor-Navigated I-124-PET/US Fusion Imaging versus Conventional Diagnostics for Retrospective Functional Assessment of Thyroid Nodules by Medical Students

**DOI:** 10.3390/s20123409

**Published:** 2020-06-17

**Authors:** Martin Freesmeyer, Thomas Winkens, Luis Weissenrieder, Christian Kühnel, Falk Gühne, Simone Schenke, Robert Drescher, Philipp Seifert

**Affiliations:** 1Clinic of Nuclear Medicine, Jena University Hospital, D-07749 Jena, Germany; Thomas.Winkens@med.uni-jena.de (T.W.); Luis.weissenrieder@gmx.net (L.W.); christian.kuehnel@med.uni-jena.de (C.K.); falk.guehne@med.uni-jena.de (F.G.); robert.drescher@med.uni-jena.de (R.D.); philipp.seifert@med.uni-jena.de (P.S.); 2Division of Nuclear Medicine, Department of Radiology and Nuclear Medicine, Magdeburg University Hospital, D-39120 Magdeburg, Germany; simone.schenke@med.ovgu.de

**Keywords:** ultrasound, multimodal imaging, sensor-navigated fusion imaging, thyroid nodules, iodine-124, positron emission tomography, medical students

## Abstract

In conventional thyroid diagnostics, the topographical correlation between thyroid nodules (TN) depicted on ultrasound (US) in axial or sagittal orientation and coronally displayed scintigraphy images can be challenging. Sensor-navigated I-124-PET/US fusion imaging has been introduced as a problem-solving tool for ambiguous cases. The purpose of this study was to investigate the results of multiple unexperienced medical students (MS) versus multiple nuclear medicine physicians (MD) regarding the overvalue of I-124-PET/US in comparison to conventional diagnostics (CD) for the functional assessment of TN. Methods: Out of clinical routine, cases with ambiguous findings on CD were selected for I-124-PET/US fusion imaging. Sixty-eight digital patient case files (PCF) of 34 patients (CDonly and CD+PET/US PCF) comprising 66 TN were provided to be retrospectively evaluated by 70 MD and 70 MS, respectively. A total of 2174 ratings (32.9 per TN) were carried out: 555 ratings (8.4 per TN) for CDonly and 532 ratings (8.1 per TN) for CD+PET/US by each MD and MS. Results: Functional assessment revealed 8.5%/11.7% (n.s.) (16.4%/25.8% (*p* = 0.0002)), 41.8%/28.5% (*p* < 0.0001) (23.9%/17.9% (*p* = 0.0193)), 36.0%/30.5% (n.s.) (57.3%/53.9% (n.s.)), and 13.7%/29.4% (*p* < 0.0001) (2.4%/2.4% (n.s.)) hyperfunctioning, indifferent, hypofunctioning, and not rateable TNs for CDonly (CD+PET/US) and MD/MS, respectively. The respective rating confidence was indicated as absolute certain, quite certain, equivocal, uncertain, and not rateable in 11.7/3.4% (*p* < 0.0001) (44.9%/38.9% (*p* = 0.0541), 51.9%/26.7% (*p* < 0.0001) (46.2%/41.5% (n.s.)), 21.6%/29.0% (*p* = 0.0051) (6.2%/14.8% (*p* < 0.0001)), 1.1%/11.5% (*p* < 0.0001) (0.2%/2.3% (*p* = 0.0032)), and 13.7%/29.4% (*p* < 0.0001) (2.4%/2.4% (n.s.)) by MD/MS, respectively. There was a significant difference in the diversity of the observers’ functional assessment of TN (MD 0.84 vs. MS 1.02, *p* = 0.0006) and the respective confidence in functional assessment (MD 0.93 vs. MS 1.16, *p* < 0.0001) between MD and MS on CDonly, whereas CD+PET/US revealed weaker differences for both groups (MD 0.48 vs. MS 0.47, *p* = 0.57; and MD 0.66 vs. MS 0.83, *p* = 0.0437). With the additional application of I-124-PET/US, the rating diversity of both MD and MS markedly tends towards more consistency (*p* < 0.0001 in each case). Conclusion: The additional application of sensor-navigated I-124-PET/US fusion imaging significantly influenced the functional assessment of TN positively, especially for unexperienced observers.

## 1. Introduction

Large-scaled multicentre-based health studies have shown that thyroid disorders are prevalent in around a third of the examined population [1,2,3]. Most of the findings are thyroid nodules (TN) and cysts, which are clinically insignificant benign lesions in over 90% of the examined population [4]. Recent publications report a rapidly increasing incidence of thyroid cancer [5,6,7]. This trend is caused by technical developments such as high resolution ultrasound (US) devices correlated with significant changes in clinical practice recommendations by the guidelines [8,9]. Thyroid cancer related mortality did not change, suggesting that the ascending incidence may be due to overdiagnosis of thyroid disorders [5,8,10]. 

Contrarily, there is still a remarkable number of malignant neoplasms incidentally diagnosed after thyroid surgery. Up to 50% of the thyroid carcinomas are reported to be incidental detections after thyroidectomy [11,12]. These findings reveal limitations of preoperative thyroid diagnostics [13]. Consequently, remarkable research efforts have been carried out to investigate methods supporting physicians to predict whether a TN is benign or malignant. Besides the implementation of various US-based risk stratification systems such as thyroid imaging reporting and data systems (TIRADS), technical approaches such as elastography, contrast-enhanced ultrasound (CEUS) or multispectral optoacoustic tomography have been introduced [14,15,16,17,18]. 

Thyroid scintigraphy, as a long-established method, is shrinking in importance according to the latest guideline recommendations [9,19]. The value of the method is greatly underestimated since the detection of so called “hot nodules” enables a relatively certain exclusion of malignancy in these lesions. Only few cases of autonomously hyperfunctioning TN are reported in the literature, especially if hyperthyroidism coexists [20]. Recently published cases revealed benign histopathology in >95% of hot TN with intermediate cytology results [21]. Identification of autonomously hyperfunctioning tissue in lesions with suspicious US features can help to avoid unnecessary invasive interventions [22,23]. Furthermore, scintigraphy is the key investigation to prove the possibility of radioiodine therapy as an alternative to surgery for the treatment of nodular goiter disease. 

The above-mentioned strength of thyroid scintigraphy is strongly related to a correct topographical correlation with TN detected on US. Especially in case of multinodular goiter with closely neighbouring or unfavourably located lesions, the correct alignment of findings on the transversal and sagittal US images with the coronal scintigraphy scan can be challenging [11]. In order to overcome this limitation and to achieve accurate functional characterizations of TN, hybrid imaging approaches such as sensor-navigated Tc-99m-pertechnetate single photon emission computed tomography (SPECT)/US and sensor-navigated sodium iodine-124 (I-124) positron emission tomography (PET)/US have been introduced [24,25,26,27]. Particularly, I-124-PET/US proved to be superior in comparison to the conventional diagnostics (CD), comprising B-mode US and Tc-99m-pertechnetate scintigraphy, in a multi-observer study with >100 nuclear medicine physicians [28]. The results of this study showed that the confidence in functional assessment of TN increases significantly when I-124-PET/US is added to CD. Because the correct correlation between US and scintigraphy can be demanding and is related to the clinical experience of the investigator, the authors believe that the novel method of I-124-PET/US enables any observer to gain certainty in the correct functional assessment of TN and that unexperienced observers may particularly benefit from this approach. 

The purpose of the current study is to re-evaluate the results of the experts in comparison to medical students by means of corresponding and matched analyses of their respective functional assessment of TN on CD versus sensor-navigated I-124-PET/US. 

## 2. Materials and Methods

### 2.1. Patients, Ethics, and Registrations

Patient data of a previous study, named “**F**usion **U**ltra**S**ound **I**maging **o**f the Thyroid Gla**N**d with **I**-124 PET—**E**valuation of **N**odule **A**llocation. (**FUSION iENA**)” were further evaluated [28]. Adult patients who were referred to a university hospital for clinical routine thyroid diagnostics between February 2015 and February 2017 and, who gave informed consent for additional I-124-PET/US fusion imaging in case of uncertain findings in conventional diagnostics (CD) and for the usage of their data for future research, were included. The study has been approved by the local ethics committee (Reference number: 4286-12/14), was conducted in accordance with the principles of the Declaration of Helsinki, and was registered at “www.ClinicalTrials.gov” (NCT03128255). 

### 2.2. Conventional Diagnostics (CD) and I-124-PET/US Fusion Imaging (PET/US)

CD (comprising laboratory parameters, B-mode US, and Tc-99m-pertechnetate scintigraphy) were evaluated by three experienced investigators (T.W. and P.S. have 6 years, M.F. 21 years of professional experience in thyroid imaging). In case of uncertainties on CD such as TN depicted on US with ambiguous function on scintigraphy or multinodular goiter with hyper- and hypo-functioning areas that cannot be assigned to a specific nodule on US, additional I-124-PET/US fusion imaging was performed. Patient exclusion criteria were hyperthyroidism (TSH-level < 0.25 mU/l), previous thyroid surgery or radioiodine therapy, thyroid medication or iodine exposition within the last 6 months, Tc-99m-pertechnetate uptake < 0.5%, insufficient US conditions and predominantly cystic TN > 2 cm. For sensor-navigated I-124-PET/CT, acquisition of a single bed position (PET scan time 10 min) with a low-dose cervical CT scan (Biograph mCT40; Siemens, Erlangen, Germany) was conducted approximately 24 h after oral application of 1 MBq I-124. 

Subsequently, DICOM data were transferred to the LOGIQ E9 US device (GE Healthcare, Milwaukee, WI, USA) and real-time PET/US fusion imaging using a magnetic field-based sensor-navigation system and the VNAV software (GE Healthcare) was performed. Further technical and procedural details of I-124-PET/US fusion imaging are described in several previously published papers [26,29,30,31].

### 2.3. Patient Case Files (PCF) and Observers

Patient cases were selected in accordance with the inclusion and exclusion criteria and presented in anonymized digital patient case files (PCF), which were created using dedicated presentation software (Prezi Inc., Version 4.2.1, San Francisco, CA, USA). For each patient case, two PCF were created: either comprising CD (CDonly) or both CD and PET/US (CD+PET/US). On six (CDonly) to ten (CD+PET/US) slides, all relevant clinical information were given including age, gender, clinical history, symptoms, laboratory parameters as well as the images of US, Tc-99m-pertechnetate scintigraphy and I-124PET/US. The clinical images were presented as cine loops and the observers could interactively start, stop, repeat and review all examinations. The TN selected for evaluation were clearly marked within the images. In advance, a video tutorial was provided. No time limit was given. 

After examination of the PCF, the observers were asked to rate the functional assessment of each marked TN according to a 4-point scale (hypofunctioning, indifferent, hyperfunctioning or not rateable). Furthermore, their confidence in functional assessment was asked to be rated on a 5-point scale (absolute certain, quite certain, equivocal, uncertain, not rateable).

All PCF have been presented to 106 nuclear medicine physicians (medical doctors, MD) in a previous study [28]. In this investigation, each MD rated 7.2 ± 1.8 (range: 4–14, median: 8) randomly assigned PCF. For the current study, the results of the 70 MD which rated exactly 8 PCF, were paired randomly one-to-one for the evaluation by 70 medical students (MS). The purpose of this methodology was to enable the comparison of the results of the nuclear medicine experts with those of unexperienced observers.

### 2.4. Data Analyses/Statistics

All data were recorded in Microsoft Excel software (Microsoft Corporation, Version 14.7.3, Redmond, WA, USA). Statistical analyses and graphics (Table 1 and Table 2, Figure 1, Figure 2 and Figure 3) were calculated with R software (Prozess und Statistik, Günter Faes, Version 17.11.2018, Dormagen, Germany). The visualizations of Figure 1 and Figure 3 were manually programmed analogous to the predecessor study of Winkens et al. in 2019 [28]. For the evaluation of differences between CDonly and CD+PET/US, chi-squared test with Yates’ correction or Fisher’s exact test (in case of n < 5) were used (Table 1 and Table 2); for diversity calculations, unpaired nonparametric Wilcoxon rank sum test was used (Figure 3). A *p*-value of <0.05 was considered significant for all tests. No adjustments for multiple testing was applied.

## 3. Results

### 3.1. Patient and Observer Data

Sixty-eight PCF (34 CDonly and 34 CD+PET/US) comprising 34 patients (23 female and 11 male) aged 56 ±14 years (range: 32–84, median: 53) were created. A total of 66 TN ≥ 1 cm (= 1.94 TN per patient) were included and marked in the respective PCF. Every observer (MD and MS) rated exactly 8 PCF. The MD (14 residents and 56 senior residents) had 14 ± 8 years of clinical experience (range: 1–27, median: 10). The MS were in the second phase of medical studies (5th–10th semester) and novices in the field of nuclear medicine imaging. A total of 555 ratings (8.4 per TN) were recorded for CDonly and 532 ratings (8.1 per TN) for CD+PET/US. These numbers refer to both MD and MS, so a total of 2174 ratings (32.9 per TN) were carried out.

### 3.2. Functional Assessment

On CDonly (CD+PET/US) 8.5% (16.4%), 41.8% (23.9%), 36.0% (57.3%), and 13.7% (2.4%) of the TN were rated as hyperfunctioning, indifferent, hypofunctioning, and not rateable by MD, respectively (*p* < 0.0001 for all values). MS rated 11.7% (25.8%), 28.5% (17.9%), 30.5% (53.9%), 29.4% (2.4%) of the TN as hyperfunctioning, indifferent, hypofunctioning, and not rateable, respectively, on CDonly (CD+PET/US) (*p* < 0.0001 for all values) (Table 1). Thus, highly significant more hyper- and hypo-functioning TN as well as less indifferent TN were assessed on CD+PET/US in comparison to CDonly by both MD and MS. The number of not rateable TN decreased when I-124-PET/US was added to CD, especially in the MS group. The highest rating alterations between CDonly and CD+PET/US were observed for indifferent TN on CDonly shifting to hypofunctioning TN on CD+PET/US in MD ratings, and for not rateable or indifferent TN on CDonly shifting to hypofunctioning TN on CD+PET/US in MS ratings (Figure 1).

Except for the hyperfunctioning TN (CDonly *p* = 0.0902; CD+PET/US *p* = 0.0002), a strong approximation of the rating results of MD and MS was observed when adding I-124-PET/US. This was particularly evident for the not rateable TN (CDonly *p* < 0.0001; CD+PET/US *p* = 0.99). The diversity analyses confirmed these findings. On CDonly, there was a highly significant difference in the rating diversity between MD and MS (0.84 vs. 1.02, *p* = 0.0006), whereas CD+PET/US revealed more consistent assessments for both groups (0.48 vs. 0.47, *p* = 0.57). With the additional application of I-124-PET/US to CD, the rating diversity of both MD and MS markedly tends towards more consistency (*p* < 0.0001 in each case) (Figure 3).

### 3.3. Confidence in Functional Assessment

The observers’ confidence in functional assessment increased for both MD and MS when adding I-124-PET/US to CD. MD rated the functional assessment of the TN with certainty (absolute certain + quite certain) in 63.6% of the cases on CDonly and in 91.2% on CD+PET/US, respectively (*p* < 0.0001). This improvement was even higher among MS with certain ratings (absolute certain + quite certain) in only 30.1% on CDonly versus 80.4% on CD+PET/US (*p* < 0.0001). Ambiguous ratings (equivocal + uncertain + not rateable) were carried out less often on CD+PET/US in comparison to CDonly. MD (MS) rated with ambiguities in 36.4% (69.9%) of the cases on CDonly and in 8.8% (19.6%) on CD+PET/US, respectively (*p* < 0.0001 in each case) (Table 2). In particular, the majority of the not rateable TN on CDonly could be rated with absolute or quite certainty on CD+PET/US in both MD and MS groups. A decrease in the observers’ confidence in functional assessment of TN was not observed in either MD or MS (Figure 2).

Approximations of the rating results of MD and MS were observed for certain ratings (absolute certain + quite certain) between CDonly (*p* < 0.0001) and CD+PET/US (*p* = 0.0004) as well as for the share of not rateable TN (CDonly *p* < 0.0001; CD+PET/US *p* = 0.99) (Table 2). Diversity analyses showed that there was a highly significant difference in the diversity of the observers’ confidence in functional assessment of TN between MD and MS on CDonly (0.93 vs. 1.16, *p* < 0.0001), whereas CD+PET/US revealed weaker differences for both groups (0.66 vs. 0.83, *p* = 0.0437). With the additional application of I-124-PET/US, the rating diversity of both MD and MS markedly tend towards more consistency (*p* < 0.0001 in each case) (Figure 3).

A representative example of a PCF containing CD+PET/US is demonstrated in Figure 4.

## 4. Discussion

B-mode US remains the standard method to evaluate the risk of malignancy of TN. Several US-based risk stratification systems and TIRADS provide recommendations for further clarification in terms of fine-needle aspiration cytology (FNAC) as the gold standard for preoperative diagnosis of TN [9,14,15,19,32,33]. Interobserver variability is a well-known limitation of these systems, particularly with regard to the experience of the investigators [34,35]; and their usefulness in case of coexisting diffuse thyroid disease such as Hashimoto’s thyroiditis, Graves’ disease, subacute thyroiditis or after radioiodine therapy has not been well studied so far [36]. Furthermore, it was shown that up to 80% of the scintigraphically hyperfunctioning TN should be clarified by FNAC according to the mentioned risk stratification systems [22,23]. Since scintigraphy can almost rule out malignancy in case of hot TN, the investigation has the potential to avoid unnecessary interventions [37]. Moreover, autonomously hyperfunctioning TN cannot be reliably identified or excluded by laboratory tests alone [38,39]. 

An indispensable presupposition to facilitate the strength of the scintigraphy is a correct topographical correlation with US images. An accurate functional assessment in terms of correct nodule allocation is necessary to identify proper lesions for FNAC. In that regard, the addition of hybrid approaches such as Tc-99m-pertechnetate-SPECT/US and I-124-PET/US fusion imaging has been shown to be superior to CD alone [24,25,26,27,28]. The superimposition of metabolic and morphological images in real-time enables more accurate evaluations of even small or unfavourably located TN and increases the observers’ confidence in the functional assessment of TN, especially on I-124-PET/US fusion imaging. 

### 4.1. Statement of Principal Findings

The presented data were obtained from multiple nuclear medicine specialists. Since the topographical correlation between the different spatial orientation of US and scintigraphy images requires ample experience, novices in nuclear medicine imaging may benefit even more from the novel technique. Therefore, we re-evaluated the above-mentioned study findings for medical students. 

Comparison of the one-to-one paired PCF reveals marked differences on CDonly ratings between MD and MS. In particular, almost a third of the MS could not rate the functionality of the presented TN. The experts rated such nodules more often as indifferent, which might be caused by additional consideration of the volume and tracer uptake ratios per lobe. An example for this effect is demonstrated in Figure 4. Missing clinical experience therefore seems to lead to uncertainty for MS in case of indistinguishable nodule borders, which are more likely presented in case of hyper- or hypo-functionality on scintigraphy. This thesis is supported by the fact that no significant differences between MD and MS were observed with regard to the rating of hyper- or hypo-functioning TN on CDonly. 

When I-124-PET/US was additionally presented to the observers (CD+PET/US), the number of nodules that were not rateable decreased significantly and the obtained results were exactly the same for both MD and MS. Most of the former not rateable TN were evaluated as hypofunctioning on CD+PET/US. Overall, the rates of hypo- and hyper-functioning nodules markedly increased when PET/US was added to CD, what was true for both MD and MS. These results most notably show that projective two-dimensional planar scintigraphy technology can disguise the hypofunctionality of TN, which may be present in >50%. 

A major difference between MD and MS was observed for the confidence in functional assessment on CDonly. MS were less than half as certain as MD. The proportion of absolute and quite certain ratings was >60% for MD and <30% for MS. This is clearly related to the experience level since MD had an average of 14 years of nuclear medicine practice. More interesting is the increasing concordance of both groups when PET/US was added to CD. No significant differences between MD and MS were found for absolute and quite certain ratings on CD+PET/US. On the level of small numbers, however, equivocal and uncertain ratings were carried out more frequently by MS on CD+PET/US. The distinct convergence was also found for not rateable TN and for the assessment of hypofunctioning lesions on CD+PET/US in comparison to CDonly. 

The addition of PET/US led to a significant more consistency of the ratings within the groups. Figure 3 demonstrates how both MD and MS could achieve more consistency in their ratings in all regards. Furthermore, the diversity differences between the two groups significantly decreased so that no differences in terms of functional assessment and only a slight disparity in terms of confidence in functional assessment of TN was found between MD and MS on CD+PET/US. The latter is attributed to the large gap of clinical experience. 

### 4.2. Strengths and Weaknesses of the Study 

The current results are based on multiple observer data with regard to clinically relevant patient cases processed in detail and presented via innovative PCF. The methodology appropriately allows for the statement that CD+PET/US profoundly improves the confidence in functional assessment of TN. Even for novices, PET/US mediates confidence and the gap between experts is distinctly diminished. The interobserver variability in terms of functional characterization of TN seems to be markedly lower with the additional usage of I-124-PET/US, because the ratings were carried out with significant higher consistency. However, the observers only rated 8 out of 68 PCF. To obtain reliable interobserver data, future studies need to be performed with fewer observers, but all participants should rate all cases.

The cases presented to the observers of this study are highly selective since only ambiguous CD findings were taken into account. The rates of hyper- or hypo-functioning as well as indifferent TN cannot be transferred to larger populations, but the results show that hyper- and hypo-functioning TN might be underestimated on CDonly.

The interpretation of the prepared PCF of this study may be comparable for experts and novices, but it should also be taken into account that the actual implementation of I-124-PET/US fusion imaging requires a lot of training and can only be handled by experienced operators.

The data suggest that the rate of hyper- and hypo-functioning TN is underestimated by today’s standard methodology. In addition to the above-mentioned fact that the PCF were chosen selectively from clinical routine, some of the differences in the functional assessments between CDonly and CD+PET/US may be due to the variable metabolic pathways of Tc-99m-pertechnetate and I-124. The different acquisition times of PET (24–28 h p.i.) and scintigraphy (20 min p.i.) can uncover so-called “trapping only” nodules, which occur in 1–2% of all thyroid nodules [40]. 

The usage of iodine-123 (I-123) would exclude those metabolic influence factors. I-123 is recommended over Tc-99m-pertechnetate for thyroid scintigraphy by the American Thyroid Association and the physical characteristics of I-123 are particular suitable for SPECT(/CT) and SPECT/US imaging [41]. I-123-SPECT is more widely available and established in comparison to I-124-PET/CT [42]. However, the inferiority of SPECT in comparison to PET imaging is well known [43].

### 4.3. Implications for Clinicians

Authors do not recommend the implementation of PET/US into the clinical routine as a standard examination method or to replace scintigraphy. I-124-PET/US fusion imaging is limited by the regional availability of PET/CT scanners, I-124 and suitable US devices with hybrid imaging software as well as by the personnel requirements including the skills needed for accurate fusion imaging and the time expenditure for PET/CT and PET/US examinations [27,28]. Effective whole-body doses of approximately 6.8 mSv derive from I-124-PET/CT scans [44,45].

The novel method should be considered as a problem-solving tool in cases of high clinical relevance and as an educational aid. I-124-PET/US can be used to teach beginners in nuclear medicine imaging and to reveal pitfalls for experienced physicians.

### 4.4. Unanswered Questions and Future Research

The current data must be considered carefully. Clinical suggestions cannot be carried out on the basis of this study, because histopathological results are only available of 4 out of 34 patients (6 out of 66 TN) and only 3 malignant lesions were included (classical papillary carcinomas). In 7 patients (7 TN) FNAC was performed (benign) and 1 patient (2 TN) was referred to radioiodine therapy. Until now there is no reliable histopathological or immunohistochemical gold standard for the functional characterization of TN to prove the ratings of the observers. Moreover, the results of this study need to be re-evaluated in different and larger populations. 

## 5. Conclusions

This study confirms the impact of sensor-navigated I-124-PET/US fusion imaging on the functional assessment of thyroid nodules for both nuclear medicine physicians and unexperienced medical students. The results regarding selected patient cases with ambiguous findings on conventional diagnostics show that the rate of hyper- and hypo-functioning thyroid nodules may be underestimated in clinical practice. The addition of I-124-PET/US fusion imaging to conventional diagnostics improves the confidence in functional assessment of thyroid nodules as well as the rating diversity for both experts and novices. In comparison to conventional diagnostics, significant rating concordance can be achieved between physicians and students. 

Considering personnel and structural limitations as well as the need for further prospective studies assessing clinical outcomes, sensor-navigated I-124-PET/US seems to be a promising problem-solving tool in cases of ambiguous findings when using conventional thyroid diagnostics. The method facilitates reliable functional characterizations of thyroid nodules even for unexperienced investigators.

## Figures and Tables

**Figure 1 sensors-20-03409-f001:**
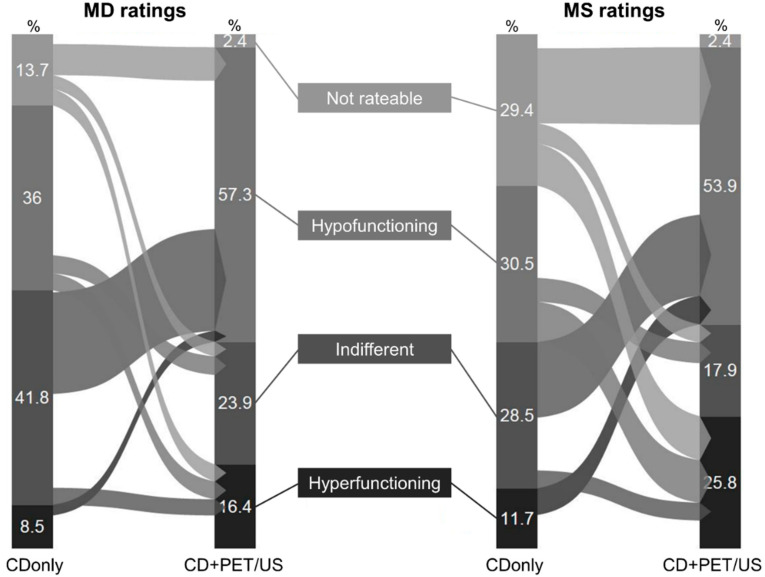
Illustration of the relative rating shifts in the functional assessment of thyroid nodules (TN) between conventional diagnostics (CDonly) and CD+I-124-PET/US fusion imaging (CD+PET/US). The left part of the graphic represents the ratings of the nuclear medicine physicians (medical doctors, MD) and the right part the ratings of the medical students (MS).

**Figure 2 sensors-20-03409-f002:**
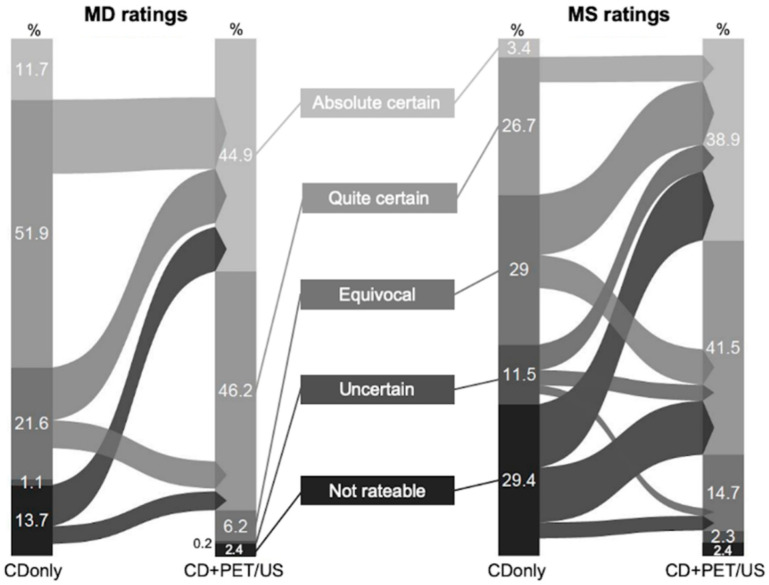
Illustration of the relative rating shifts in the observers’ confidence in functional assessment of thyroid nodules (TN) between conventional diagnostics (CDonly) and CD+I-124-PET/US fusion imaging (CD+PET/US). The left part of the graphic represents the ratings of the nuclear medicine physicians (medical doctors, MD) and the right part the ratings of the medical students (MS).

**Figure 3 sensors-20-03409-f003:**
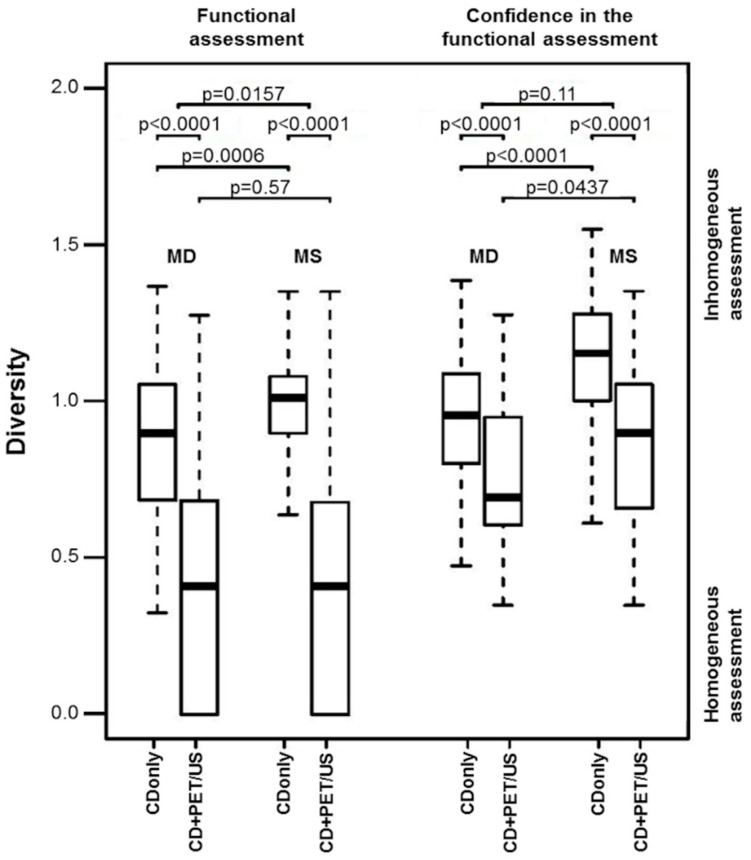
Diversity calculations of the observers’ functional assessment and the observers’ confidence in functional assessment of thyroid nodules (TN) for conventional diagnostics (CDonly) and CD+I-124-PET/US fusion imaging (CD+PET/US) illustrated by box plots. The horizontal lines within the boxes represent the median, the amplitude of each box the interquartile range, and the whiskers represent the minima and maxima. Higher diversity values represent more consistent assessments. The graphic illustrates a contrasting juxtaposition of the results of the nuclear medicine physicians (medical doctors, MD) and the medical students (MS).

**Figure 4 sensors-20-03409-f004:**
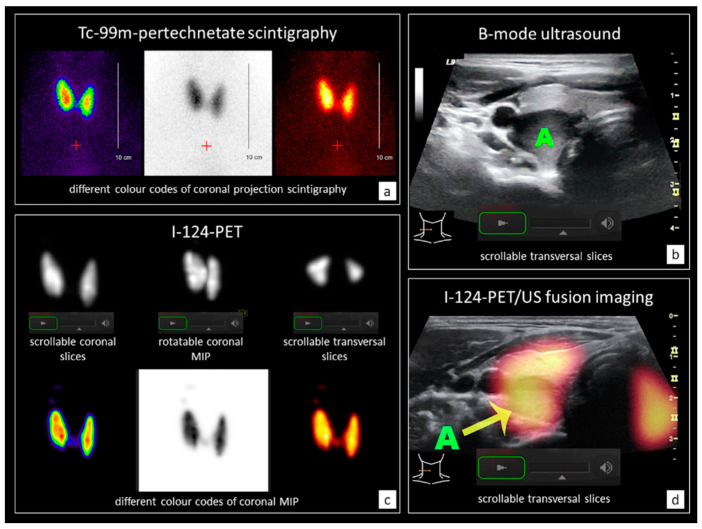
Exemplary extracts from a digital patient case file (PCF) of a 34-year old female with a single thyroid nodule (23 × 14 × 13 mm, 2.0 mL) in the right-sided thyroid lobe. (**a**) Tc-99m-pertechnetate scintigraphy in coronal orientation presented in three different colour codes (spectrum, inverted grayscale, and hot body); (**b**) Scrollable B-mode ultrasound (US) of the right lobe in transversal orientation. The solitary dorsal nodule is marked (A). (**c**) I-124-PET images of the thyroid presented in scrollable coronal and transversal slices as well as in a rotatable coronal maximum intention projection (MIP) in grayscale in the upper row. Furthermore, coronal orientated MIPs in three different colour codes (spectrum, inverted grayscale, and hot body) are presented in the lower row. (**d**) Scrollable I-124-PET/US fusion images of the right lobe in transversal orientation. The solitary dorsal nodule is marked (A). In the PCF with conventional diagnostics only (CDonly), this dorsally located nodule was prevailingly rated as indifferent with uncertain confidence by nuclear medicine physicians (medical doctors, MD) and as not rateable by medical students (MS). In the PCF with additional I-124-PET/US fusion imaging (CD+PET/US), all observers rated the nodule as indifferent with absolute or quite certainty. The entire PCF presented to the observers included further images such as transversal- and sagittal-orientated images of the whole thyroid on both B-mode US and I-124-PET/US fusion imaging, as well as further clinical data such as the thyroid volume (15 mL, right lobe: 10 mL, left lobe: 5 mL), the Tc-99m-uptake (2.02%, right lobe: 1.16%, left lobe: 0.86%), the blood levels (TSH-level: 3.6 mU/l, no pathological antibodies), and symptoms (globus sensation).

**Table 1 sensors-20-03409-t001:** Functional assessment of thyroid nodules (TN) in conventional diagnostics (CDonly) versus CD+I-124-PET/US fusion imaging (CD+PET/US); comparison between the ratings of nuclear medicine physicians (medical doctor, MD) and medical students (MS).

Functional Assessment	CDonly (n = 555)	CD+PET/US (n = 532)	*p* Values
**Hyperfunctioning, n (%)**			
MD	47 (8.5)	87 (16.4)	<0.0001
MS	65 (11.7)	137 (25.8)	<0.0001
*p* values	0.0902	0.0002	
**Indifferent, n (%)**			
MD	232 (41.8)	127 (23.9)	<0.0001
MS	158 (28.5)	95 (17.9)	<0.0001
*p* values	<0.0001	0.0193	
**Hypofunctioning, n (%)**			
MD	200 (36.0)	305 (57.3)	<0.0001
MS	169 (30.5)	287 (53.9)	<0.0001
*p* values	0.0559	0.31	
**Not Rateable, n (%)**			
MD	76 (13.7)	13 (2.4)	<0.0001
MS	163 (29.4)	13 (2.4)	<0.0001
*p* values	<0.0001	0.99	

TN—thyroid nodule; CDonly—conventional diagnostics; CD+PET/US—conventional diagnostics and I-124 positron emission tomography/ultrasonography fusion imaging; MD—medical doctors; MS—medical students; n—numbers.

**Table 2 sensors-20-03409-t002:** Confidence in the functional assessment of thyroid nodules (TN) in conventional diagnostics (CDonly) versus CD+I-124-PET/US fusion imaging (CD+PET/US); comparison between the ratings of nuclear medicine physicians (medical doctors) and medical students (MS).

Rating Confidence	CDonly (n = 555)	CD+PET/US (n = 532)	*p* Values
**Absolute Certain, n (%)**			
MD	65 (11.7)	239 (44.9)	<0.0001
MS	19 (3.4)	207 (38.9)	<0.0001
*p* values	<0.0001	0.0541	
**Quite Certain, n (%)**			
MD	288 (51.9)	246 (46.2)	0.0715
MS	148 (26.7)	221 (41.5)	<0.0001
*p* values	<0.0001	0.15	
**Equivocal, n (%)**			
MD	120 (21.6)	33 (6.2)	<0.0001
MS	161 (29.0)	79 (14.8)	<0.0001
*p* values	0.0051	<0.0001	
**Uncertain, n (%)**			
MD	6 (1.1)	1 (0.2)	0.1244
MS	64 (11.5)	12 (2.3)	<0.0001
*p* values	<0.0001	0.0032	
**Not Rateable, n (%)**			
MD	76 (13.7)	13 (2.4)	<0.0001
MS	163 (29.4)	13 (2.4)	<0.0001
*p* values	<0.0001	0.99	

TN—thyroid nodule; CDonly—conventional diagnostics; CD+PET/US—conventional diagnostics and I-124 positron emission tomography/ultrasonography fusion imaging; MD—medical doctors; MS—medical students; n—numbers.

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
