# Peer review of "Fusion iENA Scholar Study: Sensor-Navigated I-124-PET/US Fusion Imaging versus Conventional Diagnostics for Retrospective Functional Assessment of Thyroid Nodules by Medical Students"

_sensors, 2020, doi:10.3390/s20123409_

Round 1

Reviewer 1 Report

Freesmeyer M, Winkens T et al. have submitted a brilliantly conducted research that confirms the superiority of sensor-navigated I-124-PET/US fusion on the functional assessment of ambiguous thyroid nodules by comparing responses between medical students and experienced nuclear medicine physicians.

I agree with their claim that this novel imaging modality can be useful in case of ambiguous findings over conventional imaging only in experienced centers with the available tools.

I believe that this excellent work by the authors deserves the publication in your journal sensors.

Author Response

Response letter

To Reviewer1,

We thank you very much for taking the time to carefully read our manuscript. We are very grateful for your exceptional appreciation.

Best regards,

Martin Freesmeyer
On behalf of all of the authors

Reviewer 2 Report

The authors shall be congratulated to their well-written and well-defined spin-off study to their previous main study. Below, you will find mostly corrections and suggestions of minor kind, only the statistics section (and, maybe, respective results) need partly clarification (and, hence, maybe revision of respective results). The discussion section should be re-structured as proposed below or in a similar manner.

Functional assessment revealed 11.7% / 8.5% [n.s.] (16.4% / 25.8% [p=0.0002]), 41.8% / 28.5% [p<0.0001] (23.9% / 17.9% [p=0.0193]), 36.0% / 30.5% [n.s.] (57.3% / 53.9% [n.s.]), and 13.7% / 29.4% [p<0.0001] (2.4% / 2.4% [n.s.]) hyperfunctioning, indifferent, hypofunctioning, and not rateable TNs for MD / MS, respectively.

l.35 8.5% / 11.7% - it’s the other way round (MD / MS)

l.38. TNs for CD only (CD+PET/US) and MD / MS, respectively .

l.42-47 Please quantify the magnitude of the differences; p-values alone do not sufficiently indicate these

l.50 influenced (past tense)

l.50 proposal: “influenced the functional assessment of TN positively, especially…”

l.105 no full stop after “PET”, no comma before “were”

l.106 either add a comma before “who” or delete comma in l. 108

l.149 paired randomly – as any pairing (by age, sex) due to the huge difference in experience does not contribute positively anyway. Please add “randomly” if this was what you meant, thanks.

l.156-157 What does “(Prozess und Statistik, Günter Faes, Version 17.11.2018, Dormagen, 156 Germany)” mean here? Was the statistical analysis outsourced?

l.157 (thereafter) Please add a reference to the approach behind Fig.1 and Fig.3. There are predecessors in the display of thermic visualization, maybe you are able to cite a reference in the context of agreement studies?! If there is a specific R package that was used to this end, please name it, thanks.

l.159 suggests (probably) unpaired, nonparametric testing with Wilcoxon rank sum test, please clarify by specifying the applied test. In Fig.3, also within-MD and within-MS comparisons were performed. Did MDs (and MSs) perform paired ratings for CD only and CD+PET/US? If so, a paired, nonparametric test (like Wilcoxon’s signed rank test) should have been performed for these within-MD and within-MS ratings. L.221-222 do suggest that an observer both judged CD only (first) and CD+PET/US (thereafter).

l.160 Please add “No adjustment for multiple testing was applied.”

l.166 Delete comma

Table 1, Fig.3, Table 2 and at respective locations in the text: please use only two digits for p>=0.10, e.g. 0.31 or 0.99

In the text, figures appear in the order 1-3-2, please correct.

l.246-249 It’s unclear what is meant by “slight / strong approximation”. Please revise.

l.258-281 Font type changed. Is this whole text part of the legend of Fig.4?! Please reformat appropriately.

l.283-381 The discussion section would greatly benefit from applying a structure like the one proposed by Docherty and Smith in an BMJ editorial in 1999 (https://doi.org/10.1136/bmj.318.7193.1224). This will both help the authors pursuing a clear-cut line of argumentation and the readers due to higher accessibility and clarity.

Author Response

Response letter

To Reviewer2,

We thank you very much for taking the time to carefully read our manuscript and helping us to further improve it through the valuable comments you have provided. Your feedback helped us considerably to resubmit a better manuscript. We critically reviewed and corrected the manuscript and also provided suggestions for answering the requests. Please kindly notice that we tried to integrate the suggestions of 3 different reviewers. Therefore, you may find changes in the manuscript that were not suggested by you.

Please find below our detailed response to each of your comments.

Best regards,

Martin Freesmeyer
On behalf of all of the authors

Comments and Suggestions for Authors: point-by-point response

The authors shall be congratulated to their well-written and well-defined spin-off study to their previous main study. Below, you will find mostly corrections and suggestions of minor kind, only the statistics section (and, maybe, respective results) need partly clarification (and, hence, maybe revision of respective results). The discussion section should be re-structured as proposed below or in a similar manner.

Functional assessment revealed 11.7% / 8.5% [n.s.] (16.4% / 25.8% [p=0.0002]), 41.8% / 28.5% [p<0.0001] (23.9% / 17.9% [p=0.0193]), 36.0% / 30.5% [n.s.] (57.3% / 53.9% [n.s.]), and 13.7% / 29.4% [p<0.0001] (2.4% / 2.4% [n.s.]) hyperfunctioning, indifferent, hypofunctioning, and not rateable TNs for MD / MS, respectively.

l.35 8.5% / 11.7% - it’s the other way round (MD / MS)

l.38. TNs for CD only (CD+PET/US) and MD / MS, respectively .

l.42-47 Please quantify the magnitude of the differences; p-values alone do not sufficiently indicate these

l.50 influenced (past tense)

l.50 proposal: “influenced the functional assessment of TN positively, especially…”

  • Thank you very much for bringing these deficiencies to our attention. We corrected the „Abstract“ accordingly. (lines 35-51) We additionally added the requested values for diversity calculations in „Results“ (lines 219, 220, 272 and 273)

l.105 no full stop after “PET”, no comma before “were”

  • We replaced the full stop with a hyphen and deleted the comma. We hope that the revised version is appropriate. (line 110)

l.106 either add a comma before “who” or delete comma in l. 108

  • We added a comma before „who“. (line 112)

l.149 paired randomly – as any pairing (by age, sex) due to the huge difference in experience does not contribute positively anyway. Please add “randomly” if this was what you meant, thanks.

  • We thank you very much for this advice. You exactly met our point and we accordingly added „randomly“. (line 157)

l.156-157 What does “(Prozess und Statistik, Günter Faes, Version 17.11.2018, Dormagen, 156 Germany)” mean here? Was the statistical analysis outsourced?

  • Those are the customer specifics of the company that created the R software which we used for statistical calculations (they were not outsourced).

l.157 (thereafter) Please add a reference to the approach behind Fig.1 and Fig.3. There are predecessors in the display of thermic visualization, maybe you are able to cite a reference in the context of agreement studies?! If there is a specific R package that was used to this end, please name it, thanks.

  • The visualizations were manually programmed in R software by our research group. There are no specific R software packages available. The same style was previously used in the mentioned predecessor study (Winkens et al. 2019, reference #28). We accordingly added a remark and referenced the predecessor study at the requested position in “Data analyses/ statistics”. (lines 165-6)

l.159 suggests (probably) unpaired, nonparametric testing with Wilcoxon rank sum test, please clarify by specifying the applied test. In Fig.3, also within-MD and within-MS comparisons were performed. Did MDs (and MSs) perform paired ratings for CD only and CD+PET/US? If so, a paired, nonparametric test (like Wilcoxon’s signed rank test) should have been performed for these within-MD and within-MS ratings. L.221-222 do suggest that an observer both judged CD only (first) and CD+PET/US (thereafter).

  • You are right, a unpaired nonparametric testing with Wilcoxon rank sum test was applied, we specified “Data analyses/ statistics” accordingly. (line 168) The p-values of figure 3 were all calculated using the unpaired nonparametric Wilcoxon rank sum test, because the MDs and MSs did not perform paired ratings. One observer rated 8 out of 68 (34 CDonly and the respective 34 CD+PET/US) randomly assigned PCF. We mentioned that limitation in “Strengths and weaknesses of the study”.

l.160 Please add “No adjustment for multiple testing was applied.”

l.166 Delete comma

  • We changed “Data analyses/ statistics” as suggested. (lines 169-70 and line 176)

Table 1, Fig.3, Table 2 and at respective locations in the text: please use only two digits for p>=0.10, e.g. 0.31 or 0.99

  • We changed all p>=0.10 in all tables, figure 3 and in the whole manuscript accordingly.

In the text, figures appear in the order 1-3-2, please correct.

  • Thank you for that important hint, we rearranged the figures in the correct order.

l.246-249 It’s unclear what is meant by “slight / strong approximation”. Please revise.

  • We intended to rate the level of approximation by giving them attributes but the change of the p-values are demonstrating that very well so we fully agree that those words are not necessary. We changed the respective paragraph in “Confidence in functional assessment”. (lines 268-70)

l.258-281 Font type changed. Is this whole text part of the legend of Fig.4?! Please reformat appropriately.

  • We thank you for bringing this to our attention. We reformatted the legend of figure 4 appropriately. (lines 297-317)

l.283-381 The discussion section would greatly benefit from applying a structure like the one proposed by Docherty and Smith in an BMJ editorial in 1999 (https://doi.org/10.1136/bmj.318.7193.1224). This will both help the authors pursuing a clear-cut line of argumentation and the readers due to higher accessibility and clarity.

  • Thank you very much for this very valuable advice! We changed “Discussion” according to the suggested structure for discussion of scientific papers provided by Docherty and Smith in 1999.

Reviewer 3 Report

The manuscript by Freesmeyer et al compares conventional diagnostics against a hybrid PET/US fusion technique, for functional assessment of thyroid nodules. Using data from a previous study (FUSION iENA), the authors assessed the potential of sensor-based co-registration of I124-PET and US in aiding unexperienced medical students with classifying the functional state of detected nodules (i.e., hyper- vs. hypo-functioning, indifferent, and non-rateable). They found that PET/US enhances the confidence of diagnosis in both experts and medical students. Additionally, improved concordance between experts and students can be achieved with the fused modality, thereby bridging the experience gap between the two cohorts.

This clinical study builds upon previous work and confirms the benefit of using fused techniques, which combine structural and functional components.  The manuscript is written in an extremely clear manner, with defined aims and outcomes. The statistical analysis is robust and the conclusions strongly founded on the data. A limitation section is included, which is always welcome. Nevertheless, as authors acknowledge, PET/US is far from clinical practice, but could be useful as an educational tool or as a last resort in ambiguous clinical cases. Additionally, this manuscript is relevant to the clinical readership, but is of moderate importance to the technical readership of Sensors.

Nevertheless, I believe that this study has enough novelty and scientific merit. Therefore, I recommend it for publication in Sensors following minor revision.

My details comments follow:

Line 73: Authors can provide the main reasons for abandoning thyroid scintigraphy from clinical practice. Is it due to time constraints? Resolution? Radiation exposure?

Line 98: So is this a retrospective study? No new scans were performed? If so, it should be mentioned as such in the title and abstract.

Line 107: Did the inform consent or HIPAA form mention that patient data might be used in future studies?

Line 115: CD (i.e., diagnostics) were evaluated.

Line 140: By marking, do the authors mean segmented or simply indicated with an arrow? In the former case, was segmentation performed manually or automatically? It is not clear from figure 4.

Line 149: Why is one-to-one pairing necessary?

Lines 157-159: How did the authors confirm the assumptions of each statistical test?

Line 283: I wouldn't call it high resolution. What is the ultrasound frequency commonly used for thyroid examination?

Author Response

Response letter

To Reviewer3,

We thank you very much for taking the time to carefully read our manuscript and helping us to further improve it through the valuable comments you have provided. We critically reviewed and corrected the manuscript and also provided suggestions for answering the requests. Please kindly notice that we tried to integrate the suggestions of 3 different reviewers. Therefore, you may find changes in the manuscript that were not suggested by you.

Please find below our detailed response to each of your comments.

Best regards,

Martin Freesmeyer
On behalf of all of the authors

Comments and Suggestions for Authors: point-by-point response

The manuscript by Freesmeyer et al compares conventional diagnostics against a hybrid PET/US fusion technique, for functional assessment of thyroid nodules. Using data from a previous study (FUSION iENA), the authors assessed the potential of sensor-based co-registration of I124-PET and US in aiding unexperienced medical students with classifying the functional state of detected nodules (i.e., hyper- vs. hypo-functioning, indifferent, and non-rateable). They found that PET/US enhances the confidence of diagnosis in both experts and medical students. Additionally, improved concordance between experts and students can be achieved with the fused modality, thereby bridging the experience gap between the two cohorts.

This clinical study builds upon previous work and confirms the benefit of using fused techniques, which combine structural and functional components.  The manuscript is written in an extremely clear manner, with defined aims and outcomes. The statistical analysis is robust and the conclusions strongly founded on the data. A limitation section is included, which is always welcome. Nevertheless, as authors acknowledge, PET/US is far from clinical practice, but could be useful as an educational tool or as a last resort in ambiguous clinical cases. Additionally, this manuscript is relevant to the clinical readership, but is of moderate importance to the technical readership of Sensors.

Nevertheless, I believe that this study has enough novelty and scientific merit. Therefore, I recommend it for publication in Sensors following minor revision.

My details comments follow:

Line 73: Authors can provide the main reasons for abandoning thyroid scintigraphy from clinical practice. Is it due to time constraints? Resolution? Radiation exposure?

  • Thank you very much for bringing this question regarding clinical developments to our attention. We don’t experience abandoning of thyroid scintigraph from the clinical practice in our region (Germany). The investigation is still frequently requested at our sites. The study rather addresses the limitations of this methodology regarding the topographical assignment of TNs between US and scintigraphy. The majority of the patients is appropriately diagnosed with this clinical standard (conventional diagnostics), but in some cases of unfavorable located TNs or ambiguous scintigraphic images more detailed imaging is necessary to reach certainty in the functional assessment. Therefore, we introduced I-124-PET/US as an problem solving tool and evaluated it for clinically selected patient cases. Time expenditure and radiation exposure of this novel approach are higher in comparison to thyroid scintigraphy and therefore we don’t recommend replacing the standard scintigraphy. However, the resolution and diagnostic accuracy of I-124-PET/US is superior.

Line 98: So is this a retrospective study? No new scans were performed? If so, it should be mentioned as such in the title and abstract.

  • We added the retrospective study design to the title and abstract. (lines 4 and 31)

Line 107: Did the inform consent or HIPAA form mention that patient data might be used in future studies?

  • Yes, every patient prospectively signed a dedicated informed consent form that included information regarding the possible use of their data for future research. We added a respective note to “Patients, ethics and registrations”. (lines 113-4)

Line 115: CD (i.e., diagnostics) were evaluated.

  • We changed “was” to “were”. (line 120)

Line 140: By marking, do the authors mean segmented or simply indicated with an arrow? In the former case, was segmentation performed manually or automatically? It is not clear from figure 4.

  • By marking we mean that we clearly marked the relevant TNs so that the observers can unambiguously recognize which nodule is intended for ratings. We did this with letters (big letter “A” in figure 4b and 4d). In case of I-124-PET/US we additionally used arrows to ensure that the PET images superimposed to the US images are not covered by the big letters. We did not perform any kind of segmentation.

Line 149: Why is one-to-one pairing necessary?

  • This methodology is not indispensable necessary, but it appeared to be the easiest way to ensure that the exact same PCFs were rated by both MD and MS.

Lines 157-159: How did the authors confirm the assumptions of each statistical test?

  • The statistical tests were choosen with respect to the clinical question and the used methodology. We considered p<0.05 to reveal significant differences between the respective groups. We precised the section “Data analyses/ statistics”. (lines 165-70)

Line 283: I wouldn't call it high resolution. What is the ultrasound frequency commonly used for thyroid examination?

  • The nowadays commonly used frequencies for thyroid ultrasound are 10-15 MHz which is called “high resolution” US in the literature since the older devices were not able to provide that high frequencies. We deleted “with high resolution devices” as suggested. (line 321)
